# Effect of Rootstock Genotype and Arbuscular Mycorrhizal Fungal (AMF) Species on Early Colonization of Apple

**DOI:** 10.3390/plants13101388

**Published:** 2024-05-16

**Authors:** Chris Cook, David Huskey, Mark Mazzola, Tracey Somera

**Affiliations:** 1Tree Fruit Research and Extension Center, Washington State University, 1100 N Western Ave, Wenatchee, WA 98801, USA; chris.j.cook@wsu.edu; 2United States Department of Agriculture-Agricultural Research Service Tree Fruit Research Lab, 1104 N Western Ave, Wenatchee, WA 98801, USA; 3Department of Plant Pathology, Stellenbosch University, Private Bag X1, Matieland 7600, South Africa; mark@resourcesolutionsnow.com

**Keywords:** arbuscular mycorrhizal fungi, *Malus domestica*, rootstock genotype, colonization efficacy

## Abstract

The effect of plant cultivar on the degree of mycorrhization and the benefits mediated by arbuscular mycorrhizal fungi (AMF) have been documented in many crops. In apple, a wide variety of rootstocks are commercially available; however, it is not clear whether some rootstock genotypes are more susceptible to mycorrhization than others and/or whether AMF species identity influences rootstock compatibility. This study addresses these questions by directly testing the ability/efficacy of four different AMF species (*Rhizophagus irregularis*, *Septoglomus deserticola*, *Claroideoglomus claroideum* or *Claroideoglomus etunicatum*) to colonize a variety of commercially available Geneva apple rootstock genotypes (G.11, G.41, G.210, G.969, and G.890). Briefly, micropropagated plantlets were inoculated with individual species of AMF or were not inoculated. The effects of the rootstock genotype/AMF interaction on mycorrhization, plant growth, and/or leaf nutrient concentrations were assessed. We found that both rootstock genotype and the identity of the AMF are significant sources of variation affecting the percentage of colonization. However, these factors largely operate independently in terms of the extent of root colonization. Among the AMF tested, *C. etunicatum* and *R. irregularis* represented the most compatible fungal partners, regardless of apple rootstock genotype. Among the rootstocks tested, semi-dwarfing rootstocks appeared to have an advantage over dwarfing rootstocks in regard to establishing and maintaining associations with AMF. Nutrient uptake and plant growth outcomes were also influenced in a rootstock genotype/AMF species-specific manner. Our findings suggest that matching host genetics with compatible AMF species has the potential to enhance agricultural practices in nursery and orchard systems.

## 1. Introduction

Most plants, including apple, are classified as holobionts. They consist of the plant host and a diverse assemblage of microbes. Endophytic microorganisms, including arbuscular mycorrhizal fungi (AMF) play a role in promoting the health and growth of their host plant. AMF are soilborne organisms that form symbioses with ~80% of land plants and play an integral part of functioning terrestrial ecosystems, including agroecosystems [1]. AMF emerging from roots enlarge the root–soil interface and can improve a plant’s ability to access nutrients and water [2]. These types of symbioses generally involve the trade of plant photosynthate (up to 30%) in exchange for AMF-derived resources [3,4]. In addition to improving access to nutrients, AMF have been shown to provide a spectrum of other benefits to their plant partners, including increased soil aggregation [5,6] and tolerance to environmental stressors [7,8,9]. In a recent study, symbiosis with the mycorrhizal fungi *Rhizophagus irregularis* was shown to alleviate drought stress in both apple seedlings [8] and tissue-cultured plantlets cultivated in sterilized potting mix [10]. Research also suggests that mycorrhization may improve a plant’s ability to defend itself against pathogens, both above and below ground [11,12,13]. For example, colonization of apple by mycorrhizal fungi has been shown to reduce infection by root, trunk, and twig-infecting fungi, including *Rosellinia necatrix* (root rot) [14], *Botryosphaeria* sp. [15], and *Neonectria ditissima* (apple canker) [16].

Although many AMF species are “generalists”, able to colonize a wide range of host plants, the effects of plant cultivar and/or AMF species on AMF-mediated benefits can vary greatly. This has been documented in a variety of annual crops including tomato [17], maize [18], and strawberry [19,20], as well as in perennial fruit trees including peach, plum, cherry [21], apple [22,23], and fig [24]. In apple, harnessing plant/AMF interactions to promote plant health has been shown to be effective in many situations. Notably, in one study with apple seedlings, it was found that colonization by *Funneliformis mosseae* (but not *Glomus microcarpus*) resulted in growth increases (as measured by shoot dry weight) equivalent to 400–800 mg of phosphorus fertilizer per kg of “natural” soil obtained from forested land at the edge of an orchard [25]. In a more recent study, micropropagated Murabakaido plantlets (the main apple rootstock used in Brazil) benefited from inoculation with several AMF species (*Acaulospora colombiana*, *Acaulospora morrowiae*, *Claroideoglomus etunicatum*, *Gigaspora albida*) and had increased water use efficiency and nutrient content under a variety of nutrient and moisture regimes [26]. One AMF species (*G. albida*) yielded these benefits more than the others.

Rootstocks are used for a variety of reasons, including tree size control, precocity, and disease/pest resistance. Apple rootstock genotype has been shown to be an important factor influencing the structural composition [27,28,29,30] and functional capabilities [31,32] of the root-associated microbiome. The ability of apple to suppress root infection by soil-borne pathogens depends, in part, on rootstock genotype. However, it is not yet clear whether some rootstock genotypes are more prone to AMF mycorrhization than others. In a recent study, in which six different apple rootstock genotypes were cultivated in two different replant orchard soil types, G.890 consistently harboured the highest relative percentage of AMF species [29]. G.890 showed a consistently high degree of association with Glomeraceae (as a family), especially *F. mosseae* in two different soil types. By comparison, some species of AMF (e.g., *Serendipita vermifera*, *R. irregularis*) were consistently found in greater relative abundance in the fungal communities of replant-susceptible Malling rootstocks [29], suggesting that certain rootstock genotypes may favour association with certain AMF species.

Both dwarfing and semi-dwarfing rootstocks are used for size control in high-density fruit production systems. Rootstock vigor may be another factor influencing the ability of a tree to form relationships with AMF due to differences in carbon and/or resource (e.g., phosphate) use partitioning. For example, a recent study characterized the effect of rootstock vigor on plant–water relations as well as the subsequent impact on net CO_2_ assimilation [33]; net leaf carbon assimilation rates were lower in rootstocks with lower vigor (i.e., more dwarfing rootstocks). Consequently, rootstock dwarfing capacity may influence carbon allocation toward mycorrhizal roots. In addition, Leisso et al. compared root exudate profiles between two dwarfing apple rootstocks (G.935 and M.26) and reported that glucose was 2× higher in root exudates from G.935 [34]. Further, research indicates that glucose may be preferred by AMF over sucrose or fructose [35,36].

In addition to host genetics, the outcome of AMF mycorrhization is likely to depend on complex interactions between environmental conditions and/or other soil/rhizosphere microorganisms [37]. In apple, replant disease is caused by multiple soil-borne pathogens which impede the establishment of new plantings (of apple or closely related species) on the same site. Although AMF occupy the same ecological niche (the root cortex) as soil-borne plant pathogens, a root colonized by mycorrhizal fungi is unlikely to be simultaneously colonized by non-mycorrhizal fungi and vice versa [38]. For example, Caruso et al. observed higher levels of AMF colonization in roots of field-grown healthy apple trees compared to replant-affected tress [24]. In the Van Horn et al. study mentioned above, experiments were also conducted in replant soil (in which soil-borne pathogens may restrict the degree of mycorrhization) [29]. To disentangle host effects from possible effects of soil-type and other synergistic and/or antagonistic interactions with soil-borne microbes (which may inadvertently influence the capacity of rootstocks to form associations with AMF), all experiments from the study reported here were conducted using micropropagated plantlets in pasteurized planting media, conditions largely designed to minimize the influence of microbes other than the AMF inoculum.

Although there is evidence suggesting that rootstock–fungi association preferences exist in apple, studies which directly examine the influence of both apple rootstock genotype and AMF species on the efficacy of mycorrhization are limited. Our study was largely designed to answer the following questions: (1) do certain apple rootstock genotypes have a greater proclivity for mycorrhization than others? and (2) is rootstock receptivity AMF species-specific? Assessment of plant growth and leaf nutrient concentrations were also evaluated; however, these results were secondary to assessing the specificity of the AMF–rootstock association.

We hypothesized that some apple rootstock cultivars are generally more receptive to mycorrhization (or are more dependent on AMF) than others and that certain AM fungal species will preferentially colonize certain rootstocks. Ascertaining the specificity of apple-rootstock–AMF combinations that influence tree growth is of importance for the application of management practices that are employed to improve tree establishment at planting. Studies addressing this issue are particularly relevant due to the broad availability of commercial mycorrhizal inoculants that have not been verified to form associations with apple or improve tree growth. Alternatively, if rootstock genotype and/or AMF species have little influence on percent colonization, the implication may be that Geneva apple rootstock cultivars have broad receptivity for multiple species of AMF.

## 2. Materials and Methods

Selection and evaluation of mycorrhizal inoculum: the AMF species used in this experiment belong to two different families within the order Glomerales: Glomeraceae *(R. irregularis* and *Septoglomus deserticola*) and Claroideoglomeraceae (*Claroideoglomus claroideum* and *C. etunicatum*) [39]. These AMF were considered “ecologically relevant” because they represent species previously documented in apple roots and/or rhizospheres [13,26,29,40,41]. Single species inocula were obtained from Mycointech (Tarragona, Spain). Spores contained in each inoculum were isolated using the sucrose gradient centrifugation technique as described by the International Collection of Vesicular Arbuscular Mycorrhizal Fungi (INVAM) (https://invam.ku.edu/spore-extraction (accessed on 16 February 2021)). Spore viability and confirmation of species identity was then evaluated microscopically under the guidance of Bill Wheeler (INVAM curator) (Appendix A). Species confirmation was also attempted via lysing single spores according to the INVAM protocol “DNA Extraction from Single Spores” (https://invam.ku.edu/dna-analysis (accessed on 6 October 2022)). PCR amplification of DNA originating from single spores was conducted using the Glomeromycota-specific primer set AML1/2 [42]. PCR conditions were as follows: 94 °C for 5 min, 40 cycles of amplification consisting of 30 s at 94 °C, 1 min at 60 °C, and 1 min at 72 °C, and a final extension at 72 °C for 5 min. PCR-purified DNA was sequenced using the same primers (Eton Bioscience Inc., San Diego, CA, USA). Sequences were trimmed in Sequence Scanner v2.0 (Applied Biosystems, Waltham, MA, USA) and identified using the MaarjAM database (https://maarjam.ut.ee/ (accessed on 6 October 2022)) (Appendix A).

Initial planting: micropropagated dwarfing (G.11, G.41) and semi-dwarfing (G.210, G.890, and G.969) apple rootstock genotypes were received as plantlets from North American Plants, Inc. (McMinnville, OR, USA). G.11 and G.41 are closely related by pedigree, as are G.890 and G.969 [43]. G.11 and G.41 resulted from crossing ‘Robusta 5’ with M.26 and M.27, respectively. G.890 and G.969 are progenies from the cross of ‘Ottawa 3’ by ‘Robusta 5’. Potting mix (Sunshine Professional Growing Mix #1; Sun Gro^®^ Horticulture, Abbotsford, BC, Canada) was pasteurized at 80 °C for 2 × 8 h cycles, with a cool down and aseptic mixing step in between heating cycles. Individual plantlets were immediately planted into pasteurized potting mix in 1 L pots. Growth conditions were 22.5 °C with a 16 h/8 h light/dark cycle, with the photosynthetic photon flux density (PPFD) maintained at 70–80 µmol/m^2^/s. After six (Exp 2) or eight (Exp 1) weeks of growth, plantlets were large enough for use in experiments. During this time, powdery mildew, aphid, and spider mite infestations were topically treated with Procure 50 WS (Chemtura, Middlebury, CT, USA), MilStop (BioWorks, Victor, NY, USA), Rally 40 WSP (Corteva Agroscience, Indianapolis, IN, USA), and Dr. Earth Insect Killer (Dr. Earth, Inc., Winters, CA, USA), respectively.

Experimental design and planting: two separate experiments of similar design were conducted to assess compatibility of rootstock genotype × AMF combinations. The first experiment (Exp 1), conducted in 2021, was repeated in 2023 (Exp 2) due to low levels of AMF colonization. Each experiment included 16 different rootstock genotype × AMF combinations (four rootstock genotypes × four AMF species) with a “no AMF” control treatment for each rootstock genotype. In Exp 1, a randomized block layout containing 20 different treatments × 7 replicate pots each (140 plants) was used. In Exp 2, a fully randomized design was used for each of three different block timepoints (2-, 5- or 8-weeks post-inoculation), with three replicate pots for each treatment × timepoint combination. The semi-dwarfing rootstock G.210 was not available for use in Exp 2; therefore, G.969 (also semi-dwarfing) was used instead. For each rootstock genotype, plantlets of similar size were selected. Prior to planting, the root systems of all plants were briefly rinsed in water to remove attached growth medium and manipulated by hand to expose more of the root system to the AMF formulation. Root volume was measured by submersing roots in water and by measuring the volume of water displaced in mL, and total plant fresh weight was measured using an electronic balance (Mettler PC 2000, Mettler-Toledo LLC, Columbus, OH, USA).

The AMF inoculant consisted of a granular formulation and was incorporated into the root zone prior to planting by sprinkling 5 g around the root ball and into the planting hole. In Exp 1, average spore concentrations were 168, 166, 173, and 197 spores/g for *R. irregularis*, *S. deserticola*, *C. claroideum*, and *C. etunicatum,* respectively (830–985 spores/plant). In Exp 2, mean spore concentrations were 98, 175, 335, and 260 spores/g for *R. irregularis*, *S. deserticola*, *C. claroideum*, and *C. etunicatum*, respectively (490–1675 spores/plant). Plants were then immediately planted into 3.7 L pots containing pasteurized potting soil (Exp 1) or pasteurized orchard soil (Exp 2). The soil used in Exp 2 was obtained from Sunrise Research Orchard, near Rock Island, WA [47.311551, −120.068531]), and was pasteurized as described above. Soil phosphorus (P) and nitrogen (N) analysis was conducted on each soil type by Soiltest Farm Consultants, Inc. (Moses Lake, WA, USA) using the Olsen and flow injection analysis methods, respectively. In both experiments, growth conditions were 22.5 °C with a 16 h/8 h light/dark cycle, PPFD was maintained at 70–80 µmol/m^2^/s, and plants were watered as needed. To reduce risk of contamination to “no AMF” control plants, pots were either placed on grid top raised platforms (Exp 1) or in individual aluminium dishes (Exp 2).

Experimental harvest: in Exp 1, plants were harvested after 5 weeks, which was considered to be an agriculturally relevant timeframe for sufficient colonization to occur [23,44]. In Exp 2, plants were harvested after 2, 5, and 8 weeks. Upon harvest, root-associated soil was removed via gentle manual manipulation and rinsing. Roots were patted dry with paper towels. Total plant fresh weight, root fresh mass (Exp 1), or root volume (Exp 2) and leader shoot length were measured. Shoot length was measured from the soil line to the tip of the meristem using a ruler. Root systems from each individual plant were spread out on a sterilized working surface and fine roots (<1.5 mm) were selected from various positions to obtain a representative sample. In Exp 1, whole root systems were loosely wrapped in moistened paper towels, placed in a sealed plastic bag, and stored at 4 °C until processing/staining (up to 3 weeks). In Exp 2, fine roots were immediately sampled from plant root systems at harvest and cut into 1–2 cm lengths. Pre-processed roots were then stored in wet paper towels at 4 °C until staining (~1 week). The remaining root tissue was frozen at −80 °C. In both experiments, leaf tissue was also collected at harvest and stored in brown paper bags prior to oven drying (80 °C, 48 h).

Leaf tissue analysis: leaf tissue nutrient analyses for total nitrogen and total phosphorus were conducted by Soiltest Farm Consultants, Inc. (Moses Lake, WA, USA). In Exp 1, because there was insufficient leaf tissue for analysis of individual plants, three technical replicates were analyzed from each treatment which consisted of leaf tissue pooled from multiple plants within the same treatment. In Exp 2, the limited number of biological replicates available across the multiple sampling timepoints did not allow for sufficient plant material to conduct leaf nutrient analysis.

Quantification of AMF root colonization: from each plant, approximately 2 g of fine-root tissue was stained following a modified protocol by Phillips and Hayman (1970) [45] adapted from INVAM and briefly described here. Root sections were placed in plastic tissue biopsy cassettes (Micro mesh, Ted Pella, Inc., Redding, CA, USA) and cleared by immersion in KOH (10% *w*/*v*) for 90 min in a 95 °C water bath. Cassettes were then rinsed with 5× full changes of DI water. Cleared, rinsed roots were acidified by soaking for 1 hr in room temperature HCl (2% *v*/*v*). Acidified roots were stained for 1 hr in a staining solution of either 0.05% direct blue (Exp 1) or trypan blue (Exp 2) in lactoglycerol (1:1:1, *v*:*v*:*v*, water: glycerol: lactic acid) in a 95 °C water bath. After staining, cassettes were removed from the stain and rinsed with 5× full changes of DI water. Prior to microscopic examination of root tissues, cassettes were stored at 4 °C in DI water for a maximum of 1–2 months. In Exp 1, the degree of mycorrhization for each rootstock × AMF combination was determined under a stereo microscope using the classical gridline intersect method of Giovannetti and Mosse [46]. All 140 samples were examined separately. In Exp 2, the McGonigle et al. [47] gridline intersect for degree of mycorrhization was used. Stained root segments were arranged lengthwise in rows on a microscope slide containing a polyvinyl–lacto–glycerol (PVLG) mount and gently crushed with a cover slip. All samples were analyzed separately and there were three technical replicate slides per plant. The degree of mycorrhization for each rootstock × AMF combination was determined using a compound microscope (Olympus BX53, Olympus Bartlett, TN, USA). The presence/absence of stained fungal hyphae, arbuscules, or vesicles was recorded at 40× or 100× magnification for >100 fields of view (Figure 1). Percent colonization of AMF was calculated as (# of fields of view with AMF present)/(total # of fields of view examined) × 100.

Statistical analyses: GraphPad Prism 10.0.2 (GraphPad Software, San Diego, CA, USA) was used to analyse the experimental data. Data from Exp 1 and Exp 2 were analyzed separately. To test the overall effect of inoculation on the percentage of AMF colonization, colonization data were transformed (Y = sqrt(Y)) prior to analysis and normality was confirmed using q–q plot assessment and the Shapiro–Wilk test after transformation (*p* value > 0.05). Transformed data were then compared using an unpaired *t*-test. Next, transformed/normalized (Y = sqrt(Y)) percent colonization data were used in a two-way ANOVA to test whether rootstock genotype, AMF species type, and their interaction had a significant influence on percent colonization. The main effects of rootstock genotype and AMF treatment on colonization were separately assessed using the Kruskal–Wallis test followed by Dunn’s multiple comparisons test. To assess differences in mean colonization between treatments for a given rootstock genotype, Welch’s ANOVA test followed by Dunnett’s T3 multiple comparisons test was used. When comparing percent colonization data between dwarfing (G.11 and G.41) and semi-dwarfing (G.969 and G.890) rootstocks, the Mann–Whitney U test was used.

Prior to analyses, all plant growth data were transformed (y = log(y)) and normality was confirmed via q–q plot analysis and Shapiro–Wilk tests. In both experiments, the effects of rootstock genotype, AMF species, and their interaction on plant growth characteristics (5 weeks post-inoculation) were assessed via a two-way ANOVA followed by Tukey’s multiple comparisons test. The overall effects of AMF inoculation on plant growth characteristics were assessed for each rootstock genotype using unpaired *t*-tests (inoculated vs. no-AMF control). For Exp 2, at each timepoint, the effects of specific rootstock genotype/AMF combinations on plant growth were assessed via the Kruskal–Wallis test followed by Dunn’s multiple comparisons test.

For Exp 1, a two-way ANOVA followed by Tukey’s multiple comparisons test was also used to assess the effects of apple rootstock genotype and AMF species on leaf nutrient content. Leaf nutrient data were normal and were not transformed prior to analysis. All treatment combinations tested passed both Shapiro–Wilk and Kolmogorov–Smirnov normality tests (*p* value > 0.05). Simple linear regression analysis was used to explore the relationship between total leaf foliar Nitrogen (% of dry weight) and leader shoot length (cm).

## 3. Results

Effects of rootstock genotype and AMF species on root colonization: AMF colonization was lower in Exp 1 (0–17%) than in Exp 2 (0–57%) (Table 1; 5 weeks post-inoculation). This was most likely due to differences in soil phosphorus levels between the two experiments. After Exp 1 was conducted, we learned that the pasteurized potting soil used contained an excessive amounts of plant-available N (181 mg/kg) and P (186 mg/kg). Therefore, the experiment was repeated using pasteurized orchard soil containing a much lower levels of N (20 mg/kg) and P (15 mg/kg).

Although AMF colonization was low in Exp 1 (0–17%), the effect of AMF inoculation on % root colonization (5 weeks post-inoculation) was statistically significant in both experiments (unpaired *t*-test; p_Exp 1_ = 0.005 and p_Exp 2_ = <0.0001). However, neither rootstock genotype nor AMF species had a significant effect on % colonization in Exp 1. As shown in Table 1, the root tissue of uninoculated control plants was not completely free from infection in Exp 1. Roots of plants produced by micropropagation are not usually colonized by AMF and it is possible that hyphae of nonmycorrhizal fungi were categorized as mycorrhizal (positive counts included stained hyphae, vesicles, and arbuscules at the intersection of root and gridline). When the experiment was repeated in pasteurized orchard soil (low P), the effects of both AMF species and rootstock genotype on % AMF colonization were highly significant, with 33% and 21% of the total variation coming from the main effects of these two factors, respectively (Table 2; 5 weeks post-inoculation). Together, AMF inoculation and rootstock genotype explained over 50% of the variation in AMF colonization; however, no statistically significant interaction between them was detected (*p* = 0.51; 5 weeks post-inoculation). Therefore, further analyses of each of the main effects were conducted.

Regarding the overall main effect of AMF species on root colonization (Exp 2), by 5 weeks post-inoculation, percent colonization by *R. irregularis* (*p* = 0.01) and *C. etunicatum* (*p* < 0.0001) was significantly greater than that of the no-AMF control treatment; root colonization by *C. claroideum* was almost significantly greater (*p* = 0.08). In contrast, there was no difference between *S. deserticola* and the no-AMF control (*p* > 0.99). Along these same lines, *R. irregularis* (*p* = 0.04) and *C. etunicatum* (*p* < 0.0006) resulted in significantly higher rates of colonization than *S. deserticola*, while *C. claroideum* was not significantly different from any other treatment.

In terms of the main effect of rootstock genotype on percent AMF colonization, although a significant effect was identified (*p* = 0.04; Kruskal–Wallis test), adjusted *p*-values from Dunn’s multiple comparisons test were not significant. When rootstocks were compared according to vigor class, colonization rates at 5 weeks post-inoculation (Exp 2) were significantly lower in semi-dwarfing (G.969, G.890) than in dwarfing (G.11, G.41) rootstocks (G.969, G.890) (Mann–Whitney U test; *p* = 0.0087). At this timepoint, G.11 and G.41 × *C. etunicatum* and G.11 × *C. claroideum* resulted in the highest levels of mycorrhization (Table 1). When experimental treatments were directly compared within each rootstock genotype, G.11 × *C. etunicatum* showed significantly higher mean colonization than G.11 × *S. deserticola* (Table 1; *p* = 0.02). G.11 × *C. etunicatum* was also the only rootstock/AMF combination identified as being significantly different from the associated no-AMF control treatment (Table 1; *p* = 0.04). It is important to note, however, that all no-AMF control treatments were completely free from infection at this time (Table 1).

In terms of initial colonization (i.e., 2 weeks post-inoculation; Exp 2), *R. irregularis* and *C. etunicatum* appeared to colonize apple roots more readily than *S. deserticola* or *C. claroideum*. This was true for all rootstock genotypes apart from G.11 (Figure 2). By two weeks post-inoculation, mean percent colonization by *R. irregularis* and *C. etunicatum* was in the range of approximately 5–10% and 2–10%, respectively. At this time, the highest colonization rates were observed in G.969 × *C. etunicatum* (9.60%) and G.969 × *R. irregularis* (9.56%) (Table 1). Overall, mean colonization rates at 2 weeks post-inoculation were not significantly higher in semi-dwarfing (G.969, G.890 = 3.7%) than in dwarfing rootstocks (G.11, G.41 = 0.9%) (Mann–Whitney U test; *p* = 0.2). However, percent colonization by *C. etunicatum* was significantly higher in G.969 than in both dwarfing rootstocks (G.11 and G.41) but not in G.890 (Appendix A).

By 5 weeks post-inoculation, *R. irregularis* and *C. etunicatum* remained the best colonizers of apple roots overall (Figure 2). In addition, *C. claroideum* had successfully colonized all rootstock genotypes (6–25%) except for G.969 (0%). In comparison, little to no colonization by *S. deserticola* was observed for all rootstock genotypes. At this time, the highest mean colonization rates were observed in G.11 × *C. etunicatum* (41%), G.41 × *C. etunicatum* (29%), and G.11 × *C. claroideum* (25%) (Figure 2). Percent colonization by *C. etunicatum* was significantly higher in G.11 than in both semi-dwarfing rootstocks (G.890 and G.969), but not in G.41 (Appendix A). This result was surprising, considering that little to no colonization was observed in G.11 at the 2 weeks timepoint (Figure 2).

At the 8 weeks timepoint, AMF colonization was detected in the no-AMF control treatment for G.969. Therefore, only G.890 and G.11 were included in the 8-week analysis. By this time, *S. deserticola* had successfully colonized G.890 and G.969. In addition, colonization rates detected in G.890 × *C. etunicatum* (43%) and G.890 × *C. claroideum* (37%) were among the highest observed over the course of the study and were significantly higher than those of other AMF species (Table 1). In comparison, little to no colonization was detected within G.11 roots despite the high levels of colonization observed at 5 weeks post-inoculation in similarly treated plants.

Effects of AMF inoculation on plant growth: in Exp 1, the effects of rootstock genotype (but not AMF species) on plant growth characteristics were highly significant, accounting for 86%, 59%, and 54% of the variation in leader length, total fresh mass, and root fresh mass, respectively (*p* < 0.0001 in all cases; two-way ANOVA). Although the main effect of AMF treatment was not significant in Exp 1, a significant interaction (*p* = 0.02) was detected between rootstock genotype and AMF treatment for leader shoot length data. This interaction was largely explained by the differential effects of AMF species on leader shoot length in G.210. A significant difference in leader shoot length was identified between *R. irregularis* and *S. deserticola* for G.210 (Appendix A; *p* = 0.02; Tukey’s multiple comparison test), in which *S. deserticola* resulted in significantly longer leader length than *R. irregularis* (Appendix A). It should be noted, however, that in many short-term studies, there is a lack of a clear relationship between the degree of AMF mycorrhization and alterations in plant growth [48,49].

In the second trial (Exp 2), the effects of rootstock genotype on plant growth characteristics were also highly significant, accounting for 67%, 54%, and 38% of the variation in leader length, root volume, and total fresh mass, respectively (*p* < 0.0001 in all cases; two-way ANOVA). The main effect of AMF treatment on plant fresh weight was also significant, accounting for 12% of the variation (*p* = 0.03). Unlike Exp 1, significant interaction effects between the two factors were not identified for any plant growth characteristics.

The overall effect of AMF inoculation on leader shoot length was not significant for any rootstock genotype in either experiment (Table 3). In terms of total biomass, AMF inoculation did not appear to benefit plants. In Exp 1, 5 weeks post-inoculation, there was a significant reduction in the average total fresh mass of AMF-inoculated G.11 plants (relative to no-AMF controls) by approximately 6 g. Root fresh mass was also significantly lower in AMF-inoculated G.11 plants (Table 3). In Exp 2, there was a significant reduction in the average total fresh mass of G.890 plants due to AMF inoculation by approximately 13 g (Table 3). This decrease in plant mass was largely driven by a reduction in average root volume by approximately 8 g (albeit not significant; *p* = 0.1). However, this reduction in plant fresh weight (relative to no-AMF control plants) became less pronounced in G.890 × *C. claroideum* and *C. etunicatum* treatments between 5 weeks and 8 weeks, as percent AMF colonization by *Claroideoglomus* spp. increased in G.890. During this timeframe, there was an increase in G.890 plant biomass in both *Claridoglomus* treatments, but growth was negative in all other G.890 treatments (Figure 3 and Figure 4A). Along these same lines, it is worth noting that, by 8 weeks, many of the G.890 plants had become pot-bound, with the roots of some plants growing out of the pots. Thus, the continued growth of G.890 in *Claroideoglomus* treatments (between 5 weeks and 8 weeks) may have resulted from plant growth being limited by these AMF between 2 weeks and 5 weeks.

In terms of the effects of specific rootstock genotype/AMF combinations on plant growth over time (Exp 2), no significant differences in plant fresh weight were detected between the no-AMF control and any of the specific treatments at either 2, 5, or 8 weeks post-inoculation (Figure 3). It is noteworthy, however, that G.11 × *R. irregularis* and G.11 × *S. deserticola* exhibited larger increases in plant growth between 2 weeks and 5 weeks post-inoculation than the no-AMF control or either *Claroideoglomus* species (Figure 4B). Between 5 weeks and 8 weeks, however, there was a marked reduction in AMF colonization of G.11, regardless of species (Figure 2). This reduction/loss of AMF was especially notable for *C. claroideum* and *C. etunicatum*, which had exhibited 5-week colonization rates of 25% and 40%, respectively. In particular, the loss of *C. etunicatum* from G.11 between 5 weeks and 8 weeks was associated with a relatively large increase in plant biomass during this time (Figure 3).

Effects of inoculation with AMF on foliar nutrient concentrations (Exp 1 only): the limited number of biological replicates available across the multiple sampling timepoints in Exp 2 did not allow for the collection of sufficient plant material to conduct leaf nutrient analysis. In Exp 1, although supplemental nutrients were not employed, soil phosphorus levels were extremely high (186 mg/kg). Therefore, it was not surprising that leaf P contents were relatively high (0.29–1.02%) (Appendix A). Although apple rootstock genotypes differ in their ability to take up soil nutrients and other trace elements [50], levels of P in apple leaf tissue above 0.1–0.2% of the dry weight are generally indicative of adequate phosphorus nutrition [51,52]. The combined effects of AMF treatment (3% of the variation) and rootstock genotype (68% of the variation) on foliar P content were found to be significant (*p* = 0.02). Although AMF colonization rates were relatively low in Exp 1 (8.5% = maximum), rootstock genotype and AMF treatment were found to significantly interact with each other, explaining an additional 7% of the variation (*p* = 0.02). Compared to the other rootstock genotypes, foliar P levels were relatively high in G.11, regardless of treatment. G.11 inoculated with *C. etunicatum* exhibited significantly lower foliar P content relative to that of plants inoculated with *R. irregularis* (*p* = 0.01) or relative to the uninoculated controls (*p* = 0.007) (Appendix A). In comparison, G.41 plants inoculated with *R. irregularis* had significantly lower leaf P content than uninoculated plants (*p* = 0.02).

Leaf N levels ranged from 0.99–2.76%. In apple, leaf N values above 2% are generally considered to be sufficient (pers. comm. Dr. Lee Kalcsits). It should be noted that neither P nor N concentrations were significantly correlated with the degree of mycorrhization. As observed for foliar P, both rootstock genotype and AMF treatment were found to be significant factors contributing to leaf N levels (*p* < 0.0001 for both factors; two-way ANOVA), representing 43% and 16% of the variation, respectively. In addition, rootstock genotype and AMF treatment were found to significantly interact with each other, explaining 16% of the variation (*p* = 0.001). Inoculation of G.41 and G.210 with *R. irregularis* led to a significant increase in total leaf N relative to no-AMF control plants (Figure 5). This is an intriguing finding, as leaf P content of G.41 rootstock inoculated with *R. irregularis* was significantly lower than that of the no-AMF controls (Appendix A and Figure 5). In comparison, G.11 plants inoculated with *C. claroideum* contained significantly less leaf N than those inoculated with *R. irregularis* or the no-AMF control plants. Leaf N concentrations in G.890 were significantly lower in plants inoculated with *C. claroideum* or *C. etunicatum* than with *R. irregularis* (a similar trend was observed in G.41) (Figure 5). In G.210, leaf N concentrations were significantly lower in plants inoculated with *S. deserticola* compared to those inoculated with either *R. irregularis* or *C. claroideum*.

As mentioned above, the interaction between rootstock genotype and AMF species was found to be a significant source of variation affecting leader shoot length (*p* = 0.02). This interaction was largely explained by the differential effects of AMF species on leader shoot length in G.210. For this reason, the association between leader shoot length, foliar N content, and AMF species in G.210 was further explored. As shown in Figure 6, leader shoot length was strongly negatively correlated with total leaf N in G.210 depending on which species of AMF was used as the inoculum (y = −2.17x + 10.25; r^2^ = 0.85). Total leaf N values ranged from 1.3% with *S. deserticola* to 2.1% with *R. irregularis*. Simply stated, the more nitrogen in plant leaves, the shorter the shoot.

## 4. Discussion

This study was largely designed to test whether some apple rootstock genotypes are more susceptible to mycorrhization than others and whether particular species of AMF influence rootstock compatibility. It was hypothesized that some apple rootstock cultivars are more receptive to mycorrhization than others and that specific AMF preferentially colonize certain rootstocks. Plant growth and leaf nutrient concentrations were also measured, but this assessment was secondary to evaluating the specificity of AMF-rootstock associations.

The findings from Exp 2 support the hypothesis and indicate that optimal mycorrhizal colonization of apple root systems does occur in a rootstock genotype/AMF species-specific manner. By 5 weeks post-inoculation, the main effects of both AMF species type and rootstock genotype were highly significant, accounting for over 50% of the total variation in the percentage of mycorrhization. Of that percentage, AMF species type accounted for greater variation (36%) than rootstock genotype (21%). The lack of a significant interaction between these two variables suggests that differences in percent colonization between the different AMF species are generally consistent, regardless of apple rootstock variety (and vice versa).

The AMF selected for this study are representative of species previously documented in apple roots and/or rhizospheres and were expected to interact successfully with apple. Our results show that, even though Geneva apple rootstocks can be rapidly colonized by a broad range of AMF species, some fungi are better colonizers than others. Overall, by 5 weeks post-inoculation, root colonization by *C. etunicatum* and *R. irregularis* was significantly greater than that by *S. deserticola*. In terms of initial colonization (i.e., 2 weeks post-inoculation), *C. etunicatum* and *R. irregularis* also tended to colonize apple roots more readily than *S. deserticola* or *C. claroideum.* Among the AMF tested, *C. etunicatum* and *R. irregularis* represented the most compatible fungal partners, regardless of apple rootstock genotype. In comparison, *S. deserticola* appeared to be least compatible.

Additionally, this work evaluated the differential capacity of apple rootstock genotypes to establish relationships with AMF. It was found that certain Geneva rootstocks were clearly associated with higher levels of AMF colonization than others. These association patterns did, however, change over time. For example, initially, *C. etunicatum* appeared to colonize the semi-dwarfing rootstocks (G.969 and G.890) more effectively than the dwarfing rootstocks (G.41 or G.11). Between 2 weeks and 5 weeks post-inoculation, while colonization percentages generally remained unchanged in G.969 and G.890, both G.11 and G.41 became heavily colonized by *R. irregularis*, *C. etunicatum*, and *C. claroideum*. By 5 weeks post-inoculation, the amount of colonization was significantly lower in G.969 and G.890 (semi-dwarfing) than in G.11 or G.41 (dwarfing) rootstocks. At 8 weeks, however, little to no colonization was detected within G.11 plants despite the high levels of colonization observed at 5 weeks.

It is not clear why AMF declined so dramatically in G.11 roots during the 5-to-8 week timeframe. Five weeks post-inoculation, G.11 plants colonized with both *C. etunicatum* and *C. claroideum* showed a growth depression relative to no-AMF controls. Between 5 weeks and 8 weeks, however, the reduced root colonization by *C. etunicatum* and *C. claroideum* was associated with relatively large increases in plant growth, indicating a concomitant increase in available carbon. This ability of plant hosts to regulate carbon allocation to specific mycorrhizal partners has been previously documented [53,54]. Therefore, it can be speculated that the benefits of mycorrhizal colonization to G.11 did not compensate for their costs, leading to host downregulation of photosynthate supply to the roots. In general, dwarfing rootstocks grow relatively slowly, do not grow for as long during the season, and may have less leaf area as well as reduced photosynthetic capacity relative to scions on more vigorous rootstocks [55,56,57]. Thus, some dwarfing rootstocks may be highly sensitive to the carbon costs associated with AMF colonization. It should be noted that, in more natural settings, plants simultaneously associate with multiple species of AMF varying in functional benefits. When only one fungal partner is present, interaction dynamics may be less stable.

The ability of an AMF to rapidly colonize a host is likely to benefit its competitive ability over time [58]. Therefore, another goal of this study was to identify AMF species that effectively colonize commercially available apple rootstocks in an agriculturally relevant timeframe. Resendes et al. [38] showed that mycorrhizal colonization of apple roots can occur as early as 3 days after root emergence. Resendes et al. also showed that mycorrhizal fungi selectively colonize faster growing roots. Another study found that the ability to produce/expand the primary root system may be partially determined by tree vigor (or vice versa) [43]. In our study, even though micropropagated plantlets were used, the initial mean root volume of G.890 was significantly greater than that of G.41 and G.11. The initial root volume of G.969 was also significantly greater than that of G.11. No significant differences in mean root volume were observed between G.890 and G.969 (semi-dwarfing) or between G.41 and G.11 (dwarfing). Greater initial root biomass may help explain why semi-dwarfing rootstocks tended to have higher initial percent colonization by *C. etunicatum* (i.e., 2 weeks after inoculation). This, however, is an area where additional research is needed.

In Exp 1, the unexpectedly low levels of AMF colonization were most likely due to the high level of phosphorus contained in the potting soil used. Reduced AMF root colonization when phosphorus is abundant has been documented in a variety of plant species [59,60,61]. Though most noted for their ability to transport phosphorus, AMF have also been shown to transfer nitrogen to their host plant [62,63,64,65]. In this study, the mechanisms underlying AMF-mediated variability in leaf N status and shoot height in G.210 could not be determined, and it is possible that this result was a consequence of several interacting factors. For example, in apple leaves, N content is typically positively correlated with chlorophyll content [66]. Consequently, the negative relationship between leaf N level and shoot height in G.210 may be partly due to a physiological host response to increased photosynthetic capacity. In several studies, variations in plant growth and/or nutrient uptake are not directly related to the percentage of root colonized [22,23,67]. Regardless of the mechanism(s), this outcome highlights potential benefits or costs apple rootstocks may receive by forming associations with different species of AMF, even at relatively low rates of colonization. This outcome really opens up many avenues of research from the fungal side of the association. For example, do AMF species vary in their need for carbon from the host?

It is also interesting to note that, in all rootstocks except for G.11, foliar nitrogen status in the no-AMF controls appeared to be less than adequate for optimal plant growth (<2%). When *R. irregularis* was the fungal partner, significantly more N was delivered to leaf tissue relative to uninoculated control plants (in all rootstocks except G.11). This indicates that, in apple, improved leaf nitrogen status is likely a functional benefit commonly associated with *R. irregularis.* In fact, a putative high affinity NH_4_+ transporter gene has been identified in the extra-radicle mycelium of *R. irregularis* [64,68].

In terms of growth, AMF inoculation did not appear to benefit plants in either experiment (low or high P). In Exp 1 (high P), there was a significant reduction in the total plant biomass of G.11 due to the main effect of inoculation. Previous studies have documented plant growth depression caused by AMF when phosphate availability is high [60,69]. In our study, leaf P contents indicated more than adequate phosphorus nutrition regardless of rootstock genotype or treatment, although foliar P levels were particularly high in G.11 (Exp 1). This result was surprising, considering that in a previous study, leaf mineral concentrations in Golden Delicious scions grafted onto Geneva apple rootstocks indicated that G.11 and G.969 were less efficient than G.890 and G.41 at delivering phosphorus to leaf tissue [70]. In the same experiment, leaf P concentrations were significantly correlated with total plant growth (r^2^ = 0.62). In Exp 1, leaf P concentrations were not significantly correlated with total plant growth in any rootstock genotype. However, the effects of both AMF treatment (3% of the variation) and rootstock genotype (68% of the variation) on foliar P content were significant and were found to interact with each other.

To our knowledge, this is the first study to evaluate the specificity of AMF-species–rootstock associations and report the preferences. Taken together with the Van Horn et al. 2021 study [29], our results support the hypothesis that genetic variation among apple rootstocks is likely to influence the efficiency of colonization by both inoculant and indigenous sources of AMF. In addition, some AMF species appear to be more compatible with apple than others, regardless of rootstock genotype. Both rootstock genotype and AMF species influence compatibility. Despite varied factors affecting the ability of AMF to colonize their host plants (including physical, chemical, biotic, and genetic elements), the results from this investigation are highly encouraging. These preliminary findings represent a solid step towards identifying rootstock–AMF-species preferences.

Priority effects (who colonizes the root first) can have important implications for AMF ecology and the use of fungal inoculum in sustainable agriculture [71]. Nursery-derived apple rootstocks have pre-established AMF communities which could potentially be manipulated (pre-inoculated) prior to transplanting into orchards. Whether nursery-derived AMF are displaced by the existing orchard soil community once trees are established is not yet known. In addition, rootstock/AMF combinations which are not necessarily successful in the field may still be ideal for nursery settings and/or soils that are inherently low in inoculum potential (i.e., fumigated orchard soil). As an example, in a study by Forge et al. [72], pre-plant inoculation of micropropagated Ottawa 3 rootstocks with *F. mosseae* significantly increased plant dry weight and reduced root populations of *P. penetrans* relative to non-inoculated controls in fumigated (but not in replant) orchard soil. The findings from this study have important implications for situations in which the success of inoculated AMF species is management dependent (e.g., following soil fumigation). Going forward, apple rootstock genotype and AMF species, as well as soil factors (including phosphorus levels) should be considered to effectively establish (or re-establish) target communities.

Currently, approximately 90 species of AMF have been formally described [39]; however, only a limited number of species have been the subject of study. Host–AMF relationships (and their associated functional benefits) have largely been investigated using a handful of “model” species, especially those belonging to the family Glomeraceae [25,73]. In apple, a number of endophytic fungal species (e.g., *Serendipita vermifera*) known to form mycorrhizal associations [74] and detected as endophytes in roots of apple [29] have lacked examination by the research community. This demonstrates the need for more systematic studies of plant–mycorrhizal-fungi relationships which include other fungal species (especially indigenous AMF naturally occurring in orchard soil systems which may be tolerant to suppression by high N and/or P content).

It is important to point out that many fruit trees, including apple, consist of two genetically distinct parts grafted together: a rootstock and a scion. Studies have shown that scion type can influence the effects of mycorrhization in apple [16] and in other fruit trees [75]. This is not surprising considering that different apple varieties can have different nutritional requirements. Moreover, nutrient uptake capacity may vary depending on the scion/rootstock combination [76]. In one study, N and P concentrations in Honeycrisp fruit were relatively higher in G.210 or G.41 compared with G.969 or G.11 [77]. In the same study, N concentrations in Fuji fruit were higher in G.890 > G.11 > G.210. Optimizing rootstock/scion/AMF combinations to help manage sustainability and productivity in the field presents, therefore, a major challenge within AMF research.

Unlike Malling, Geneva rootstock varieties are being planted extensively (particularly in the United States) due to their resistance/tolerance to both above ground (e.g., fire blight, wooly apple aphid) and below ground diseases (e.g., crown rot, apple replant disease) [43]. Studies show that Geneva and Malling rootstock genotypes differ in terms of root exudate metabolite profiles [34,78]. To our knowledge, information on endogenous levels of strigolactones in apple roots or root exudates is not yet available. Future studies designed to further assess the role of host genotype on AMF colonization efficacy would benefit from including Malling rootstock genotypes and/or considering specific compounds present in root exudates (e.g., strigolactones, flavonoids), which have been shown to influence the efficient establishment of mycorrhizal fungal associations.

## 5. Conclusions

Investigating the ability of different apple rootstock genotypes to form associations with different AMF species is a critical step forward in making agriculture more sustainable (e.g., the potential to improve plant nutrient uptake and water use efficiency) and productive (e.g., the potential to mitigate the detrimental effects of replant disease). In this study, it was demonstrated that apple rootstock genotype and AMF species type are both important aspects to consider if growers are to optimize the function of this relationship. In addition, this study provides evidence that the ability of certain AMF to rapidly colonize a rootstock may be related to initial root biomass and/or root architecture. Ascertaining the colonization efficiency of rootstock–AMF combinations is of particular importance in the development of practices that enhance this relationship. Such studies are particularly relevant to the industry in light of the availability of numerous commercial mycorrhizal inoculants composed of species not yet verified to form associations with apple and/or improve tree growth. This study marks a significant step toward laying the groundwork for harnessing potential apple rootstock–AMF species preferences for integration into nursery and orchard management systems. Subsequent studies in this lab will be aimed at identifying the functional benefits of specific apple rootstock–AMF associations including protection against root pathogenic fungi and tolerance to water stress.

## Figures and Tables

**Figure 1 plants-13-01388-f001:**
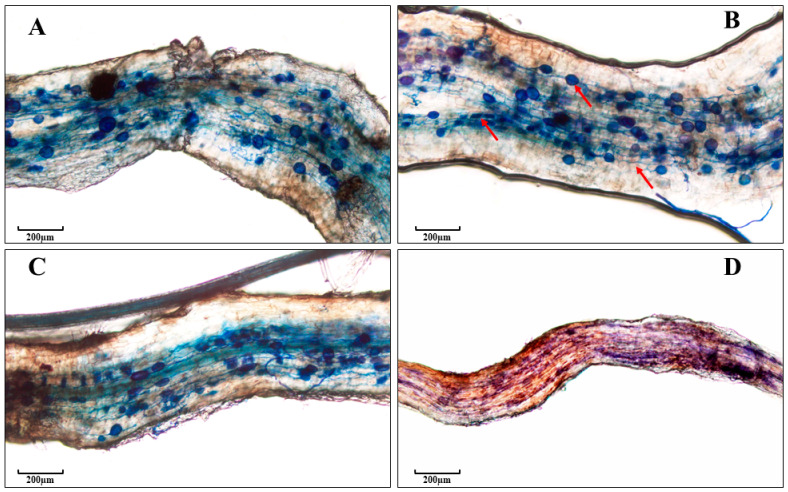
Examples of arbuscular mycorrhizal fungi in apple root tissue: *Rhizophagus irregularis* (**A**), *Claroideoglomus claroideum* (**B**), *Claroideoglomus etunicatum* (**C**), and *Septoglomus deserticola* (**D**). Roots were stained with trypan blue and viewed at 100× magnification. In (**B**), red arrows indicate representative structures (from left to right): arbuscules, vesicles, and intracellular fungal hyphae. Scale bars: 200 µm.

**Figure 2 plants-13-01388-f002:**
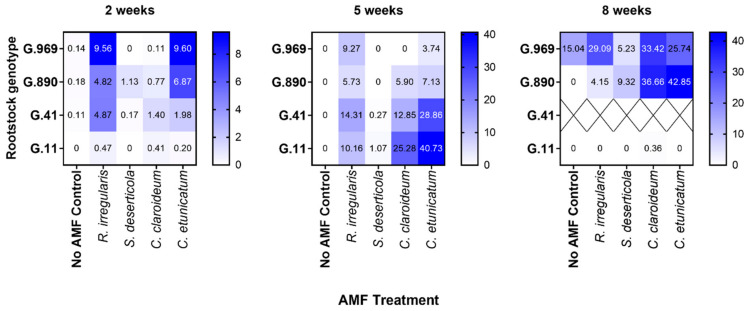
Mean percentage of root colonization in four different apple rootstock genotypes inoculated with four different species of AMF in Experiment 2. Heatmaps are shown for 2, 5, and 8 weeks post AMF inoculation, respectively. Data represent the mean of three biological replicates (× three technical replicates each) per rootstock × AMF combination per timepoint. G.41 could not be included in the 8-week analysis due to an insufficient number of plantlets.

**Figure 3 plants-13-01388-f003:**
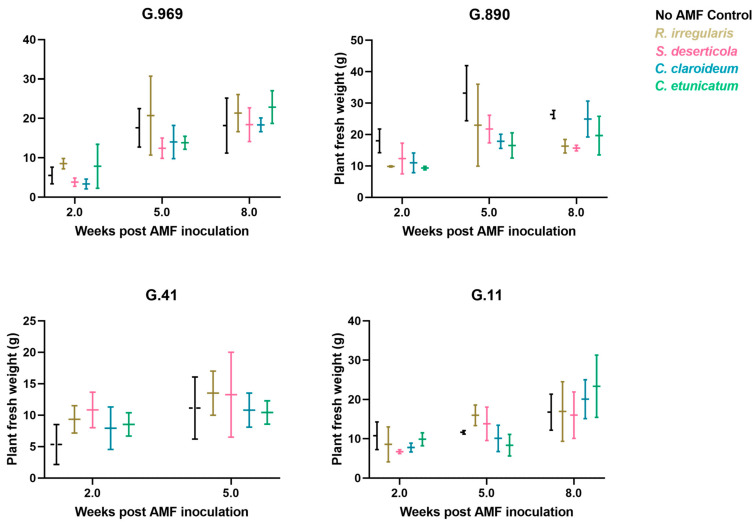
Mean plant fresh weight of apple rootstock genotypes used in Exp 2 and inoculated with four different species of AMF. Data represent the mean of three biological replicates (× three technical replicates each) per rootstock × AMF combination per timepoint. G.41 was not included in the 8-week analysis due to insufficient plantlets. Error bars show standard deviation.

**Figure 4 plants-13-01388-f004:**
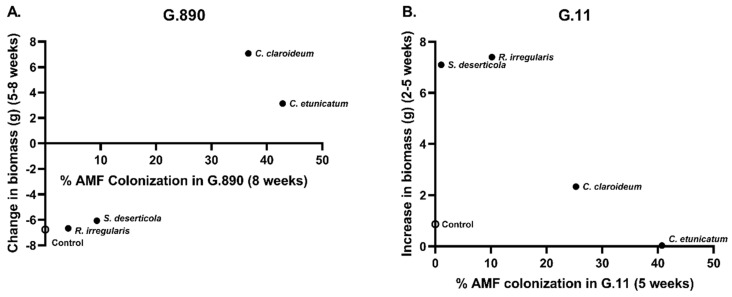
(**A**) Relationships between % AMF colonization in G.890 at 8 weeks post-inoculation and the change in plant biomass between 5 weeks and 8 weeks (Exp 2). (**B**) Relationship between % AMF colonization in G.11 at 5 weeks post-inoculation and the increase in plant biomass between 2 weeks and 5 weeks (Exp 2). Each point represents the mean of three biological replicates (×three technical replicates each).

**Figure 5 plants-13-01388-f005:**
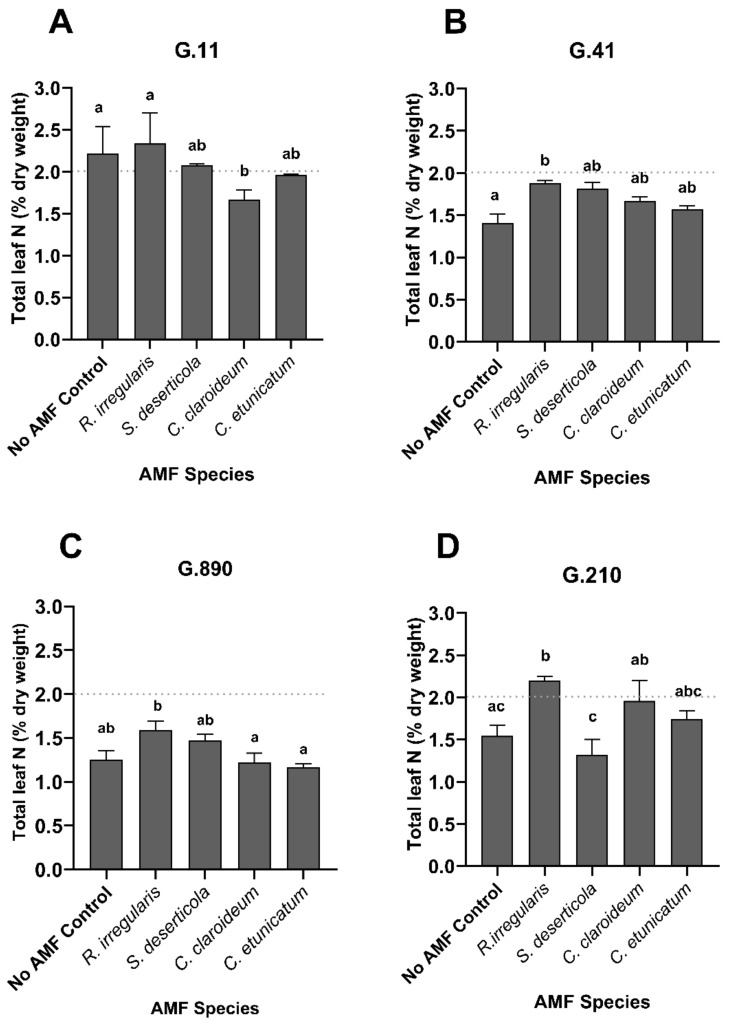
Total foliar N (% of dry weight) in the apple rootstocks (**A**) G.11, (**B**) G.41, (**C**) G.890 and (**D**) G.210 inoculated with different species of AMF, 5 weeks after inoculation in Exp 1. Letters indicate statistical differences (*p* < 0.05) in total leaf N observed according to two-way ANOVA followed by Tukey’s multiple comparisons test. The dashed horizontal line indicates adequate N nutrition (>2.0%). Data represent the mean of three technical replicates pooled from seven biological replicates.

**Figure 6 plants-13-01388-f006:**
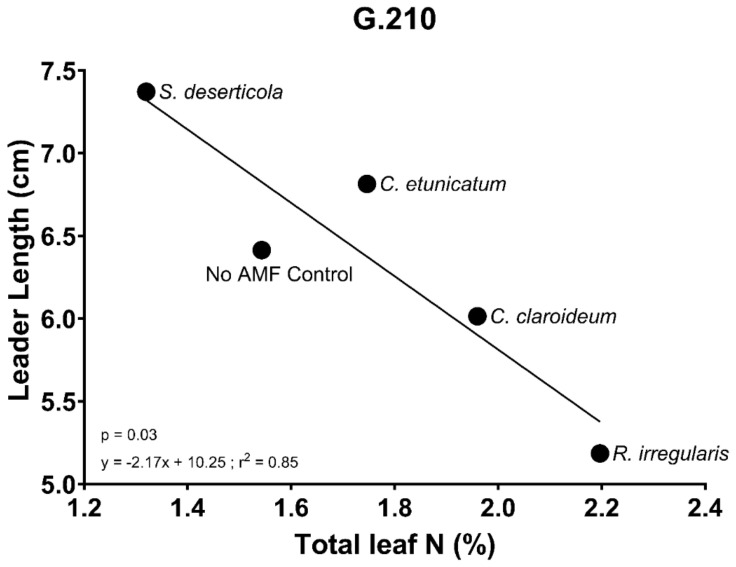
Linear regression analysis showing a significant negative correlation between total leaf nitrogen (% of dry weight) and leader shoot length (cm) in G.210 (Exp 1) depending on which species of mycorrhizal fungi the plant was inoculated with (y = −2.17x + 10.25; r^2^ = 0.85). The slope of the line is significantly non-zero (*p* = 0.03). Each point represents the mean of seven biological replicates for leader shoot length and the mean of three biological replicates for total leaf N.

**Table 1 plants-13-01388-t001:** Mean percentage of AMF colonization for each apple rootstock genotype × AMF treatment combination at 5 weeks post-inoculation in Experiment 1 and at 2, 5, and 8 weeks post-inoculation in Experiment 2. In each row, letters indicate significant differences between treatments for a given rootstock genotype (Dunnett’s T3 tests following Welch’s ANOVA). Rows without letters indicate no significant differences between any of the treatments. For simplicity, statistical differences across rootstocks for a given AMF species are shown in Appendix A. Exp 1 values are based on seven biological replicates per rootstock genotype/AMF treatment combination. Exp 2 values are based on three biological replicates (× three technical replicates each). Right and down triangle icons point in the direction of treatment and genotype information, respectively.

	Treatment	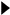	No AMF Control	*R. irregularis*	*S. deserticola*	*C. claroideum*	*C. etunicatum*
Timepoint	Genotype
Exp 15 weeks	G.11	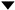	3.29	6.29	5	5	4.29
Exp 15 weeks	G.41		4.43	4.57	8.43	6.57	5.71
Exp 15 weeks	G.210		3.14	3.86	4	2.57	4.71
Exp 15 weeks	G.890		2.57	4.43	5.14	7.71	4.29
Exp 22 weeks	G.11		0	0.47	0	0.41	0.2
Exp 22 weeks	G.41		0.11 ^a^	4.87 ^b^	0.17 ^a^	1.4 ^ab^	1.98 ^b^
Exp 22 weeks	G.969		0.14 ^a^	9.56 ^ab^	0 ^a^	0.11 ^a^	9.6 ^b^
Exp 22 weeks	G.890		0.18	4.82	1.13	0.77	6.88
Exp 25 weeks	G.11		0 ^a^	10.16 ^ab^	1.07 ^a^	25.28 ^ab^	40.73 ^b^
Exp 25 weeks	G.41		0	14.31	0.27	12.85	28.86
Exp 25 weeks	G.969		0	5.73	0	0	3.74
Exp 25 weeks	G.890		0	9.27	0	5.9	7.13
Exp 28 weeks	G.11		0	0	0	0.36	0
Exp 28 weeks	G.969		15 ^ab^	29.09 ^ab^	4.59 ^a^	33.42 ^b^	25.74 ^ab^
Exp 28 weeks	G.890		0 ^a^	4.15 ^b^	9.32 ^b^	36.66 ^c^	42.85 ^c^

**Table 2 plants-13-01388-t002:** Two-way ANOVA results of the effect of rootstock genotype, AMF species, and their interaction on the percentage of AMF colonization in apple roots 5 weeks post-inoculation in Experiments 1 and 2. No-AMF controls were excluded from the analysis (AMF species treatments only). Df, degrees of freedom; Sum Sq, sum of squares; Mean Sq, mean sum of squares.

Source of Variation	Experiment	Df	Sum Sq	Mean Sq	F-Value	*p*-Value	% of Total Variation
Rootstock Genotype	Exp 1	3	3.39	1.13	1.4	0.25	3.75
AMF Species	Exp 1	3	1.75	0.58	0.72	0.54	1.94
Interaction	Exp 1	9	7.63	0.85	1.05	0.41	8.45
Residuals	Exp 1	96	77.56	0.81			
Rootstock Genotype	Exp 2	3	54.46	18.15	6.29	0.002	21.38
AMF Species	Exp 2	3	83.65	27.88	9.66	0.0001	32.85
Interaction	Exp 2	9	24.16	2.69	0.93	0.51	9.49
Residuals	Exp 2	32	92.4	2.89			

**Table 3 plants-13-01388-t003:** The overall effect of AMF inoculation on plant growth characteristics according to rootstock genotype 5 weeks post-inoculation. AMF-inoculated plants (all AMF species) were compared with non-AMF control plants using unpaired *t*-tests. Blank (light grey) spaces occur because root fresh mass and root volume were measured in Exp 1 and 2, respectively.

Rootstock	Experiment	Treatment	Leader Length (cm)	Total Fresh Mass (g)	Root Fresh Mass (g)	Root Volume (mL)
G.11	Exp 1	Inoculated	9.23	16.23	12.19	
		Control	9.74	21.91 *	17.36 *	
	Exp 2	Inoculated	11.79	12.06		6.21
		Control	12.17	11.63		7.67
G.41	Exp 1	Inoculated	19.38	30.70	20.12	
		Control	19.23	32.33	22.71	
	Exp 2	Inoculated	11.08	12.00		5.54
		Control	13.17	11.13		5.67
G.890	Exp 1	Inoculated	25.23	40.05	30.97	
		Control	21.13	46.70	37.61	
	Exp 2	Inoculated	18.38	19.78		15.63
		Control	22.00	33.17 *		23.67
G.210	Exp 1	Inoculated	6.35	17.66	13.00	
		Control	6.41	20.14	15.13	
G.969	Exp 2	Inoculated	18.96	15.23		10.38
		Control	17.67	17.60		11.00

* Significantly greater within rootstock genotype at the 0.05% level (unpaired *t*-test performed on log-transformed data).

## Data Availability

Data are contained within the article and Appendix A.

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
