# Peer review of "Effect of Rootstock Genotype and Arbuscular Mycorrhizal Fungal (AMF) Species on Early Colonization of Apple"

_plants, 2024, doi:10.3390/plants13101388_

Round 1
Reviewer 1 Report (New Reviewer)
Comments and Suggestions for Authors
The Ms by Cook et al aims at the analysis of the effect of AMF species and rootstock genotype on AMF colonization in apple. In my opinion, the manuscript is too hard to read, as it consists on the case by case description of the results of each of the different combinations used. I have two major concerns, one regarding the experiment 1, and other on the high variability of the data in Experiment 2.
A major problem is the inclusion of data from experiment 1. In this first experiment the general low level of mycorrhization was explained by the high level of phosphorus in the soil. This is probably the reason and, considering it, the authors should consider to eliminate the experiment from the manuscript, as very little information can be derived from a condition that leads to such low levels of root colonization. On top of that, in this experiment the non-inoculated treatment has colonization levels similar to those present in inoculated plants suggesting that either the pasteurization procedure did not work, or, alternatively, that thermotolerant AMF were present. Similarly, conclusions from the analysis of the effect of inoculation on plant nutrient concentration using data from this experiment might not be really relevant. The levels of P in leaf (Fig. S4) are given in % of P in plants that do have different shoot biomass. This should be taken into consideration.
Regarding experiment 2, colonization data are highly variable. There is an statistical treatment that shows some differences but there are treatments showing no significant variations in cases showing mean values of 1% vs those with 25% (G.11 at 5 weeks) or 0.27% vs 28% (G.41). The reason for such high variability is not clear, but maybe 3 biological replicates are likely not enough. In my opinion a single experiment is not enough to draw conclusions, and repeating the experiment under the same conditions is required to make sure the observed differences in compatibility between rootstocks and AMF species are consistent. In the case of 5-week time-point only one combination gives significant differences in the level of colonization as referred to the non-inoculated plants. It is hard to derive conclusions under these conditions.
Some of the conclusions are obvious. For instance, the effect of rootstock on plant growth can be anticipated from the fact that some of the dwarfing vs semi-dwarfing character of the rootstocks.
In the Discussion section some of the conclusions are expected, such as those indicating that dwarfing rootstocks have less root volume that semi-dwarfing ones.
Minor points.
The method for P determination in soil should be included. Also, it would be required to include N levels in soil
Data on Table 1 and Figure 2 are the same, on different formats (with one discrepancy, G.969/S. deserticola at 8 weeks). The heatmap hides the lack of significance of most of the differences. Data should be included only once.
Line 587: it is ref 29, not 21
The right name for P is phosphorus, not phosphorous
Fig. S3: it is confusing to include data from both experiments in the same panel. It is better to separate them, as it is the case in panels D and E.
Author Response
Please see the attachment

Reviewer 2 Report (New Reviewer)
Comments and Suggestions for Authors
The manuscript by C. Cook et al. entitled "Effect of Rootstock Genotype and Arbuscular Mycorrhizal Fungal (AMF) Species on Colonization of Apple" deals with an interesting and topical subject, namely the impact of certain Arbuscular Mycorrhizal Fungal strains on different rootstocks of the same plant species. Work of this kind could lead to major advances in the selection of future rootstocks that interact with and benefit as much as possible from Arbuscular Mycorrhizal Fungi. However, in its current version this article cannot be accepted for publication in Plants
The manuscript could be resubmitted to Plants after significant revision. I have noticed a number of major problems, listed below, which lead me to propose rejecting this article but encouraging a resubmission in regard to the interesting topic.
First of all, the authors present two trials conducted 2 years apart, and tell us that the first trial encountered a problem related to the effectiveness of colonization by Arbuscular Mycorrhizal Fungi. I don't understand the point of presenting this trial in the article. Furthermore, these two trials cannot be compared because there are differences between the rootstocks used, the measurements of physiological parameters, inoculations, growth times and the measurement of mycorrhization rates. In any case, there is too much heterogeneity between these two trials to discuss them in the same article.
Finally, it is not possible to draw any conclusions or hypotheses regarding the interaction between the Arbuscular Mycorrhizal Fungi studied and any possible impact on the growth of the various apple trees, as well as the mycorrhizal levels measured. Uninoculated apple trees are mycorrhizal. It is therefore impossible to know whether the effect observed on the inoculated apple trees is due to the Arbuscular Mycorrhizal Fungi found in the uninoculated trees, to the Arbuscular Mycorrhizal Fungi used for inoculation, or to the interaction between both types of Arbuscular Mycorrhizal Fungi. An identification of the AMF community present in the apple roots could have been a plus here.
Additional minor comments :
Lines 97-98 : Numerous studies have confirmed that arbuscular mycorrhizal fungi can strengthen the plants' systemic defense against pathogenic microbes. Despite this, information on the interaction between arbuscular mycorrhizal fungi and plant pathogens is limited. I would recommend authors to be more cautious here.
Lines 95-100 : It's hard to follow the link between the different sentences. This paragraph needs to be rewritten to make it easier to read.
Line 122 : When you talk about commercial inocula, do you mean inocula with a single strain or mixed inocula? Are you talking about the American market?
Line 125 : Please give details of the "GENEVA® apple rootstocks" if you wish to introduce them and explain why you are concentrating on them.
Lines 131-133 : “These AMF were considered “ecologically relevant” be-131 cause they represent species previously documented in apple roots and/or rhizospheres “. Why not state this in the introduction?
Lines 148-150 : How old are the rootstocks? Are they all the same age?
Line 159-160 : Was a negative control carried out without treatment to see the impact of the various treatments on Arbuscular Mycorrhizal Fungi?
Line 158 : Why 6-8 weeks ? Have you used different times? How homogenous were the plants?
Line 168 : What are the “20 different treatments” ?
Line 175 : Could you described how you have measured volume displacement ?
Line 182 : For this experiment, the difference in inoculation is really high between the different species (more than 2X). I'm not sure that this makes it possible to analyse the results. or can you justify these differences?
Line 184 : Why did you use orchard soil and not compost as in your first experiment? Did you measure the physical and chemical properties of the soil, which can have a major impact?
Line 190 : Why have you changed the method of protecting the non-inoculated control between the two experiments??
Line 192 : The mycorrhiza may have been established over such a short period of time, but is this symbiosis already functional and does it benefit the host plant at this stage? The same considerations apply to the 2 weeks of growth (Exp2).
Line 225 : Why have you changed your method of mycorrhiza quantification between the 2 experiments? Generally speaking, there are too many methodological differences between the two experiments presented.
Figure 1 : A figure showing the colonization of roots of inoculated plants for each of the 4 AMFs would be more interesting
Author Response
Please see the attachment.

Reviewer 3 Report (Previous Reviewer 2)
Comments and Suggestions for Authors
I am satisfied with the answers provided by the authors. I believe the manuscript has been significantly improved and now warrants publication in Plants.
Author Response
I am satisfied with the answers provided by the authors. I believe the manuscript has been significantly improved and now warrants publication in Plants.
Thank you for taking the time to review the many versions of the manuscript and for giving us a chance to repeat the experiment!
Reviewer 4 Report (New Reviewer)
Comments and Suggestions for Authors
The authors Cook et al. tested the hypothesis that some apple rootstock genotypes (five various genotypes) are more susceptible to mycorrhization than others. They also estimated the influence of four species of arbuscular mycorrhizal fungi in rootstock compatibility in two independent experiments. The first experiment was performed in soil with high P content, but second experiment was done in pasteurized orchard soil (low P).
The topic of this research is actual and the manuscript is generally well-written.
Minor comments
Please modify the Figure 3. At least increase the distance between different time points on the graphs (2, 5 and 8 weeks) to compare the effect.
Author Response
The authors Cook et al. tested the hypothesis that some apple rootstock genotypes (five various genotypes) are more susceptible to mycorrhization than others. They also estimated the influence of four species of arbuscular mycorrhizal fungi in rootstock compatibility in two independent experiments. The first experiment was performed in soil with high P content, but second experiment was done in pasteurized orchard soil (low P).
The topic of this research is actual and the manuscript is generally well-written.
Thank you for this compliment and for the time you have spent reviewing the manuscript.
Minor comments
Please modify the Figure 3. At least increase the distance between different time points on the graphs (2, 5 and 8 weeks) to compare the effect.
Figure 3 has been adjusted as suggested (i.e., the space between timepoints has been increased).
Round 2
Reviewer 1 Report (New Reviewer)
Comments and Suggestions for Authors
The authors have introduced some changes that have increased the quality of the MS. Out of the 6 larger comments in which my review was divided:
#1: I still think that experiment 1 should have been left out, and only briefly referred in the text with a “data not shown”. I fully agree with the fact that authors have learnt from it, but not all preliminary data are included in publications.
#2: the explanation of that non-mycorrhizal hyphae were categorized as mycorrhizal is plausible, but then, what did the authors change to avoid that and make almost 0 in Experiment 2? This should be commented.
#3: I am satisfied with the author’s response
#4: my review could not take into consideration that this was a second version of the MS with a previous review by others. Regarding experiment 2, some reasoning explaining the large degree of variability that eliminates the significance of the indicated differences should be included.
I am convinced with replies to numbers #5 and #6.
All the minor changes suggested were accepted, except in the case of that referred to ref [21]. I still maintain that [21] obviously does not correspond to Van Horn et al study, as stated in line 587 in the first version (line 624 in version with changes highlighted).
Author Response
Please see attachment

Reviewer 2 Report (New Reviewer)
Comments and Suggestions for Authors
The authors mention in the text of the article and in their answer that "“Our study was largely designed to answer the following questions: 1) Do certain apple rootstock genotypes have a greater proclivity for mycorrhization than others? and 2) Is rootstock receptivity AMF species-specific? “ Our study was not designed to assess specific functional benefits associated with certain AMF/rootstock associations; however, this is something we plan to do in a subsequent experiments.". This is an important point and should therefore be included in the title by adding "early" before "colonization of...". This point should also be mentioned in the conclusion with an opening towards functionality studies in future experiments with apple rootstocks.
Thank you to the authors for mentioning in the article that their conclusions are possible because after 2 weeks the uninoculated plants are not mycorrhized, but that it is impossible to conclude what will happen to these mycorrhizae because the same uninoculated plants are then mycorrhized.
Author Response
Please see attachment

This manuscript is a resubmission of an earlier submission. The following is a list of the peer review reports and author responses from that submission.
Round 1
Reviewer 1 Report
Comments and Suggestions for Authors
The MS has different faults. The main and most important is represented by the very low degree of root colonization, even after 5 weeks and after inoculation with a very high level of inoculum (from 166 to 197 spores/g, corresponding to 830-985 spores/plant). It is surprising that AM fungi such as Rhizophagus irregularis, Septoglomus deserticola, Claroideoglomus claroideum and Claroideoglomus etunicatum were not able to colonize the root systems at percentages higher that 9% (mean, Fig. 2). Most of the root systems showed colonization levels of 5% and lower that 5%. Moreover, Control plants showed colonization levels similar to the majority of the inoculated plants. Thus, all the results obtained are meaningless. The Authors should have tried to understand why a so high amount of spores per plant was not able to establish a good level of colonization, and why their control plants showed mycorrhizal colonization.
Reviewer 2 Report
Comments and Suggestions for Authors This study attempted to test the ability of ceratin mycorrhizal species (AMF) to effectively colonize apple plant rootstock genotypes. In this context, micropropagated seedlings were inoculated with AMF species. The main objective is to try to evaluate the effects of rootstock genotype and AMF species on root colonization, of AMF inoculation on plant growth and on foliar nutrient concentrations. Following the data, the authors argued that the nature of the AMF proves to be decisive for the degree of colonization, the nitrogen content in the leaves as well as for the development of apple tree plants. Ongoing results are encouraged. They can constitute an alternative to the exploitation of AMF species rootstocks in order to improve agricultural practices. Overall, the paper is quite well written. The methods and data are consistent and the comparison of the results obtained with those of the literature is interesting. It is work that arouses interest; it offers the possibility of developing and improving the culture of the apple tree whose socio-economic potentialities are not negligible. Nevertheless, here are some comments: • Reproducibility of experiments to validate data! • Degree of impact of factors (physico-chemical, biotic and genetic to on the effectiveness of AMF in colonizing the host plant! • Analysis of the quality of the apple tree (size, composition…)! • Inoculation Time and Period! • Reproduction of the results obtained during application in a real field!Reviewer 3 Report
Comments and Suggestions for Authors
The manuscript entitled "Effect of Rootstock Genotype and Arbuscular Mycorrhizal Fungal (AMF) Species on Colonization of Apple" shows an interesting study. This paper explored the ability/efficacy of four different AMF species to colonize different commercially available apple rootstock genotypes (G.11, G.41, G.210 and G.890). These findings provide theoretical support for matching host genetics with compatible AMF species during agricultural practices in nursery and orchard systems. This manuscript is written clearly, which can constitute the inspiration for further research also for other scientists. However, the present state of this manuscript is not sufficient for publication because of the problems of English and the details of data processing. Therefore, in my opinion, the paper needs revisions before it can be considered for publication. Some questions/suggestions are as follows:
1. Microbial names should be written regularly, for example, generic names should be italicized, (e.g., line 47: ‘Rosellinia necatrix, Botryosphaeria sp and Neonectria ditissima’ revised to ‘Rosellinia necatrix (root rot), Botryosphaeria sp. and Neonectria ditissima’), line 400, line 430 ……. Please revise it.
2. Because there was not sufficient leaf tissue for analysis of individual samples, the related analysis of leaf only consisted three biological replicates by pooling from multiple samples within the same treatment. However, what about the related analysis of root? Please provide relevant analysis details.
3. Please add the more details of pot experiments, including of the volume of pot, the properties of soil matrix, the detailed day and night temperature of planting period and soil moisture during all of planting period. Moreover, whether the seeds have been sterilized? What’s the number of seeds sown and finally retained? The related details of pot experiments are suggested to add in the manuscript. Authors may take the idea from the following references: https://doi.org/10.1016/j.ecolind.2020.106917; https://doi.org/10.1016/j.ecoenv.2021.111996.
4. The value of table1 and figure1 should be added the error value. Moreover, the number of replicates suggested to add in all of legend. For example, Each data point represents the mean of 7 biological 173 replicates.
5. English is likely the second or third language spoken by the authors. Some sentences are long-winded. I strongly recommend that the authors avail themselves of professional assistance for their writing.
6. References should be updated.
Reviewer 4 Report
Comments and Suggestions for Authors
Dear authors, the manuscript entitled "Effect of Rootstock Genotype and Arbuscular Mycorrhizal Fungal (AMF) Species on Colonization of Apple" presents and interesting approach in the field of AMF.
I consider the manuscript is well written, and each section have an appropriate length.
Overall, I have enjoyed reading the work of the authors.
Suggestions - Pay attention to the latin name of species (Abstract, Introduction etc.)
Please remove any personal expressions like we, our etc. Consider changing the sentences to look more impersonal (e.g. line 122)
Conclusion - consider rewriting this section and present the main findings of your study, with the highest or lowest results obtained.